# Heat Input and Mechanical Properties Investigation of Friction Stir Welded AA5083/AA5754 and AA5083/AA7020

**Mohamed M. Z. Ahmed [1,2,\*]**, **Sabbah Ataya [2,3]**, **Mohamed M. El-Sayed Seleman [2]**, **Abdalla M. A. Mahdy [4]**, **Naser A. Alsaleh [3]** and **Essam Ahmed [2]**

1 Department of Mechanical Engineering, College of Engineering at Al-Kharj, Prince Sattam Bin Abdulaziz University, Al Kharj 11942, Saudi Arabia
2 Department of Metallurgical and Materials Engineering, Faculty of Petroleum and Mining Engineering, Suez University, Suez 43512, Egypt; sabbah.ataya@suezuniv.edu.eg (S.A.); mohamed.elnagar@suezuniv.edu.eg (M.M.E.-S.S.); essam.ahmed@suezuniv.edu.eg (E.A.)
3 Department of Mechanical Engineering, College of Engineering, Al-Imam Mohammad Ibn Saud Islamic University, Riyadh 11432, Saudi Arabia; alsaleh@engineer.com
4 Nasr Petroleum Company, Suez 43511, Egypt; abdalla_mahdy@yahoo.com
\* Correspondence: moh.ahmed@psau.edu.sa; Tel.: +966-011-588-1200

**Abstract:** The current work presents a detailed investigation for the effect of a wide range friction stir welding (FSW) parameters on the dissimilar joints' quality of aluminum alloys. Two groups of dissimilar weldments have been produced between AA5083/AA5754 and A5083/AA7020 using tool rotational rates range from 300 to 600 rpm, and tool traverse speeds range from 20 to 80 mm/min. In addition, the effect of reversing the position of the high strength alloy at the advancing side and at retreating side has been investigated. The produced joints have been investigated using macro examination, hardness testing and tensile testing. The results showed that sound joints are obtained at the low heat input FSW parameters investigated while increasing the heat input results in tunnel defects. The hardness profile obtained in the dissimilar AA5083/AA5754 joints is the typical FSW hardness profile of these alloys in which the hardness reduced in the nugget zone due to the loss of the cold deformation strengthening. However, the profile of the dissimilar AA5083/AA7020 showed increase in the hardness in the nugget due to the intimate mixing the high strength alloy with the low strength alloy. The sound joints in both groups of the dissimilar joints showed very high joint strength with efficiency up to 97 and 98%. Having the high strength alloy at the advancing side gives high joint strength and efficiency. Furthermore, the sound joints showed ductile fracture mechanism with clear dimple features mainly and significant plastic deformation occurred before fracture. Moreover, the fracture in these joints occurred in the base materials. On the other, the joints with tunnel defect showed some features of brittle fracture due to the acceleration of the existing crack propagation upon tensile loading.

**Keywords:** friction stir welding; dissimilar welding; aluminum; mechanical properties; fracture

## 1. Introduction

AA5754 and AA5083 are aluminum magnesium alloys, and their most prominent features are the high corrosion resistance and good formability. Thus, they have been extensively used in pressure vessels, tanks, trucks and shipbuilding [1,2]. AA7020 is a precipitation-hardened aluminum alloy, demonstrating high strength per weight ratio [3]. The use of the dissimilar alloys leads to the sustainable advantages such as overall cost reduction and hybrid properties that are available in the two different alloys. Appropriate joining process and its parameters optimization plays a vital role in the service performance of these alloys. Challenges like solidification cracking, porosity, intermetallic formation and so on are present due to the difference in the chemical and physical properties of the dissimilar alloy's combinations. Recently, the friction stir welding (FSW) of dissimilar

aluminum alloys combinations has been studied extensively, which proved the potential of the process to join these alloy combinations [4–6]. However, improper FSW parameters give rise to the formation of intermetallic compounds and internal and external defects (e.g., tunnel formation, voids, surface grooves and flash) [6–10]. Therefore, the investigation of FSW parameters is very important for obtaining defect-free joints with good mechanical properties. The placement of the higher strength aluminum alloys at the advancing side (AS) or at the retreating side (RS) affects material flow as it strongly influences material the stirring and flow behavior [10,11]. This can be a crucial parameter affecting the final joint microstructure, particularly when the selected combinations of base material (BM) have significant differences in their mechanical properties, microstructure and texture [4,12–16]. Some researchers studied the effect of the placement of BM on the material flow and the resulting FSWed microstructure and the mechanical properties [17,18]. Palanivel et al. [15] revealed that the tool rotational rate and tool pin profiles affected the AA5083-H111/AA6351-T6 joint strength because of the loss of cold work in the heat affected zone (HAZ) of AA5083 side, dissolution and over-aging of precipitates of AA6351 side and macroscopic defects formation in the weld zone. Jannet and Mathews [17] concluded that the AA6061 T6/AA5083 O joints fabricated at tool rotational rate of 900 rpm yielded a higher tensile strength than those fabricated at 750 rpm contributed by the thorough plastic flow and dissolving of dissimilar alloys and due to reduction in heat generated from plastic flow of the metal at 750 rpm. Park et al. [18] showed that the materials were more properly mixed when the AA5052-H32 was in the AS and the AA6061-T6 was in the RS than the reverse case on RS. Leitao et al. [19] reported that the global mechanical behavior of the AA6016-T4/AA5182-H111 welds was a 10–20% strength reduction relative to the base materials and important losses in ductility. Khanna et al. [20] concluded that softer alloy should be placed on AS with tool offset towards it for better FSWed AA6061-T6/AA 8011-H14 qualities. Kailainathan et al. [16] showed that the tensile strength of the 6-mm-thick AA6063/AA8011 joints was increased with the increase in the tool rotational speed due to the uniform temperature distribution at the weld region. However, beyond 1200 rpm, an adverse effect was noticed due to the distortion in the weld region. Abd Elnabi et al. [21] reported that the traverse speed has the highest contribution to the process for ultimate tensile strength of AA5454/AA7075 joints. Cole et al. [22] estimated that the AA6061/AA7075 joint strength was improved with decreasing the power input to the weld because of the sensitivity of alloy to heat input and weld temperature. The work of Ouyang and Kovacevic [23] suggested that the lower-strength alloy should be placed on AS for obtaining a better weld quality. Gerard and Ehrstrom [24] mentioned that the material with the higher solidus temperature should be on the AS not only for joint quality improvement but also for internal defects/porosity elimination. Guo et al. [25] revealed that the material mixing is much more effective when AA6061 alloy was located on the AS for AA6061/AA7075 joints. The ultimate tensile strength of the joints increases with the decrease of the heat input induced by friction. Kim et al. [26] demonstrated that excessive agglomerations and defects generated by joints when the high strength Al alloy on the AS of AA5052/AA5J32 are placed due to limited flow of material. Lee et al. [27] concluded that the mechanical properties of the stir zone showed higher values when AA6061 were positioned at the RS due to the complex microstructure of the stir zone. On the other hand, Jonckheere et al. [28] showed that material flow and joint quality are more dependent on the FSW conditions and their effects on heat input and temperature distribution in weld nugget, regardless of BM placement. Due to the material plastic flow during FSW, the heat generation is controlled by tool rotation and welding speed [29–31]. However, very high rotation speeds lead to macroscopic defects because of the excessive heat input [1,32–34]. To the author's knowledge, the FSW of AA5083/AA5754 and AA5083/AA7020 have not been reported in the open literature. The present work focuses on the influences of FSW parameters more deeply including the traverse speeds (20–80 mm/min) and AS/RS positions of base materials on the quality and the mechanical properties of the dissimilar AA5083/AA5754 and AA5083/AA7020 joints.

## 2. Experimental Procedure

### 2.1. Materials

Three commercial aluminum alloys AA5083-O, AA5754-H14 and AA7020-T6 were chosen for producing dissimilar friction stir butt welds. The alloys were purchased in the form of rolled plates of 10 mm thick. The butt welds were designed to be 200 mm total width which composed of two plates; each plate was 100 mm wide and 200 mm long. The nominal chemical compositions of the parent materials are listed in Table 1. Moreover, the tensile strength, temper condition and hardness of the parent materials are summarized in Table 2.

**Table 1.** Nominal chemical composition of aluminum alloys AA5083, AA5754 and AA7020.

| Alloy | Elements in wt.% | | | | | | | | |
|---|---|---|---|---|---|---|---|---|---|
| | **Si** | **Fe** | **Cu** | **Mn** | **Mg** | **Zn** | **Cr** | **Ti** | **Al** |
| AA5083 | 0.40 | 0.40 | 0.10 | 0.4–1.0 | 4.0–4.9 | 0.25 | 0.05–0.25 | 0.15 | Bal. |
| AA5754 | 0.40 | 0.40 | 0.10 | 0.50 | 2.6–3.6 | 0.20 | 0.30 | <0.15 | Bal. |
| AA7020 | 0.35 | 0.40 | 0.20 | 0.05–0.50 | 1.0–1.4 | 4.50 | 0.1–0.35 | <0.35 | Bal. |

**Table 2.** Mechanical properties of the aluminum alloys AA5083, AA5754 and AA7020.

| Alloy | Condition | Tensile Strength, MPa | Hardness, HV |
|---|---|---|---|
| AA5083-O | Annealed | 233 | 68 |
| AA5754-H14 | Strain hardened-1/2 hard | 251 | 74 |
| AA7020-T6 | Solution heat treated and artificially aged | 364 | 117 |

### 2.2. Friction Stir Welding Procedure

The welding process was performed on the friction stir welding machine (EG-FSW-M1) at Suez University. This machine has been locally designed and manufactured in Egypt. The main motor power of this machine is 30 HP (22 kW) and can deliver torque up to 100 N·m, rotational speed up to 3000 rpm and tilt angel up to ±5°. The travel speed of the table up to 1000 mm/min. The tool design is an important parameter in FSW processes, which influences the heat generation, plastic flow, the resulting microstructure and mechanical properties of the welded material. The used rotating tool was of a cylindrical threaded pin with scrolled shoulder made of H13 tool steel that heat treated to obtain hardness of 50 HRC. The shoulder diameter was 25 mm, the pin (probe) diameter was 8 mm, and pin height was 9.8 mm, which is slightly less than the material thickness (10 mm). The angle between the edge of shoulder and the pin was 3°. The configuration of the tool used in this study is shown in Figure 1.

The hardness of the as-received tool steel was 25.3 HRC. After manufacturing the FSW tool, it has been hardened by heating to 950 °C and holding for 30 min then oil quenched, then tempered by heating to 550 °C and holding for one hour then air-cooled to room temperature. The heat treatment process was carried out using an electric resistant furnace of type Nabertherm-1200 °C. The hardness of the hardened tool steel was measured as 61 HRC. The tempering process has decreased the tool hardness to 54 HRC. Al alloy plates were prepared to obtain the required dimensions of 200 mm length and 100 mm width. The plates were clamped properly on the FSW machine table as shown in Figure 2a, b shows the butt joint after completing the FSW process.

For the system AA5083/AA5754, the plate of the alloy AA5083 was positioned in the AS, while the AA5754 plate was in the RS as illustrated in Figure 2a. Workpieces were rigidly clamped, to prevent the plates from lifting apart during the welding process. For the system AA5083/AA7020, the plate of the alloy AA5083 was positioned in the AS, while the AA7020 plate was in the RS for a set of welding conditions. For the same set of the welding conditions, the plate of the AA7020 was reversed to be positioned in the AS and

the plate of the AA5083 was in the RS, as shown in Table 3. The welding process progressed as follows: The tool was rotated and slowly plunged into the workpiece with speed of 0.1 mm/s until the shoulder of the tool forcibly contacts the upper surface of the material. After that, the tool was traversed along the weld line for a single pass weld. The tool was tilted by a constant angle of 3° against the vertical axis, so that the rear of the tool is lower than the front. This has been found to assist the forging process and the material flow during FSW. Table 3 summarizes the different combinations of operating conditions parameters investigated in this work during FSW. The desired welding parameters are based on the ongoing research at the authors laboratory in FSW of the different aluminum alloys of 10 mm thick.

### 2.3. Macrostructural Investigation

Cross sections of welded joints were prepared for metallographic analysis using standard metallographic procedure [32]. The samples were etched using Keller's reagent for a period of 40–50 s. at room temperature to reveal the macrostructure of the welded samples and then washed with water and acetone, and then air-dried.

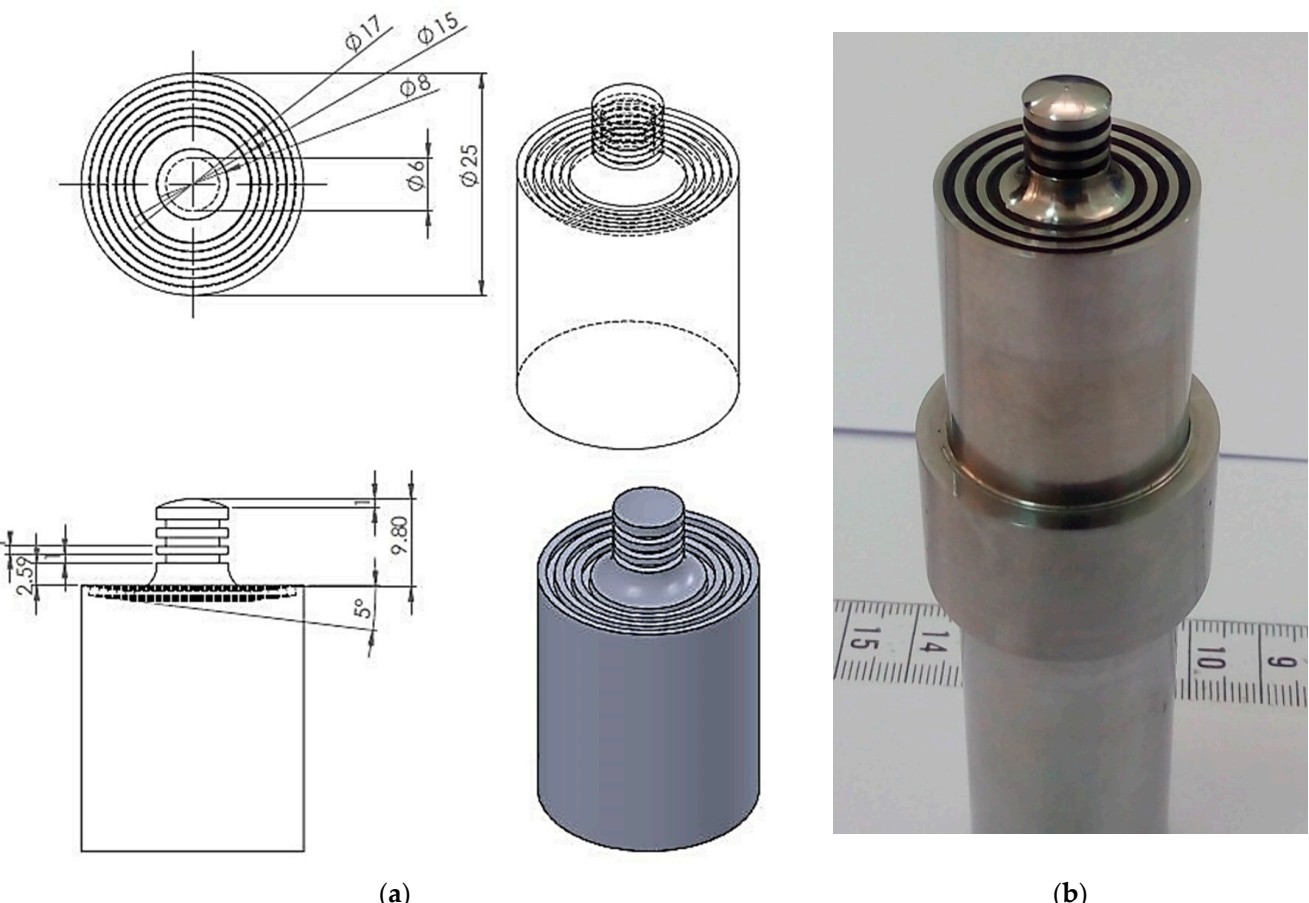

(**a**)        (**b**)

**Figure 1.** Friction stir welding (FSW) tool used in the FSW experiments (**a**) Computer aided design (CAD) drawings with detailed dimensions in mm and (**b**) image of the used FSW tool.

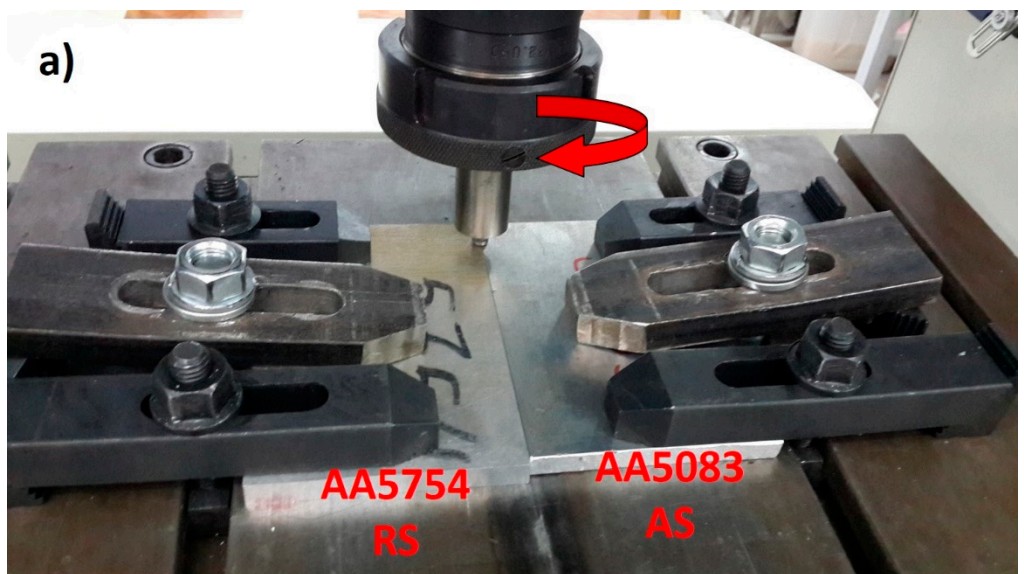

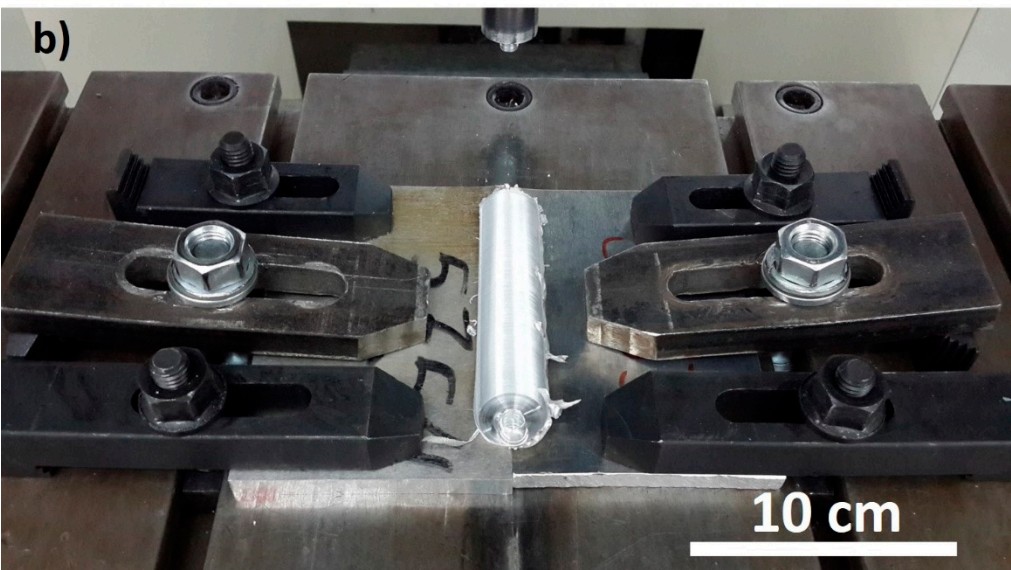

**Figure 2.** (**a**) The clamping of the plates on the FSW machine table with the advancing side (AS) and retreating side (RS) indicated and the tool rotation direction indicated also by the red arrow and (**b**) Aluminum alloys butt joint on the table after FSW.

**Table 3.** FSW welding parameters and position of alloys at the AS and RS.

| AA5083/AA5754 | | | AA5083/AA7020 | | |
|---|---|---|---|---|---|
| Rotation Speed (rpm) | Travel Speed (mm/min) | Position | Rotation Speed (rpm) | Travel Speed (mm/min) | Position |
| 400 | 20 | AA5083 AS | 500 | 20 | AA5083 AS |
|  | 40 | AA5083 AS |  | 40 | AA5083 AS |
|  | 60 | AA5083 AS |  | 80 | AA5083 AS |
| 600 | 20 | AA5083 AS | 500 | 20 | AA7020 AS |
|  | 40 | AA5083 AS |  | 40 | AA7020 AS |
|  | 60 | AA5083 AS |  | 80 | AA7020 AS |

### 2.4. Mechanical Properties

The materials were mechanically tested before and after FSW for comparison. To have an insight into the mechanical properties, hardness measurements and tensile testing were carried out. Vickers macro-hardness tests were performed on the transverse cross-sections

perpendicular to the welding direction with an interspacing distance of 2 mm using a test load of 1000 g force and dwell time of a 15 s. To evaluate the tensile properties of the welded stir zone, transverse flat tensile specimens were used. Specimens were machined perpendicular to the FSW direction to the dimensions: length of 80 mm, width of 15 mm, and thickness of 8.5 mm. The specimen's dimensions agree with the DIN EN10002-1 2001(D) standards. After machining, both surfaces of the samples were flushed to avoid any dimensional irregularity. Figure 3 shows the dimensions of the tensile specimen and an image of the sample after tensile testing. Tensile tests were carried out at room temperature with an initial crosshead speed of 0.1 mm/s using the universal testing machine Instron 4210, Norwood, MA, USA. The tensile data acquired were analyzed to determine tensile properties and joint efficiency. The fracture surface of the tension tested samples was examined using the Scanning Electron Microscope Type: Quanta 250 with a Field Emission Gun, FEI company (Hillsboro, OR, USA) to determine the failure mode of the welded samples.

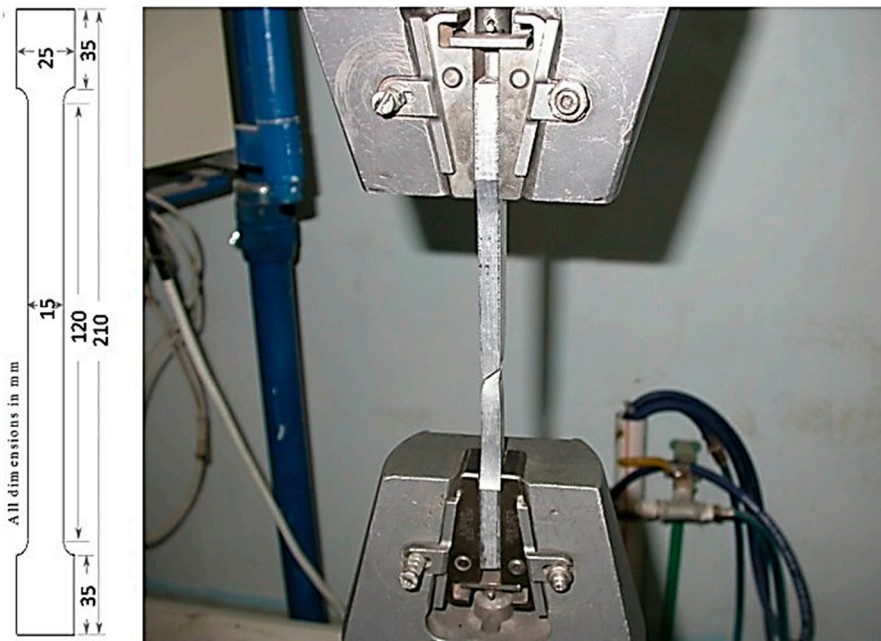

**Figure 3.** The dimensions of the tensile specimen and an image of the sample after tensile testing.

### 3. Results and Discussion

*3.1. Effect of FSW Parameters on the Heat Input*

Heat input is one of the important parameters associated with all welding processes and affects the weld quality and properties. Although, FSW is characterized by low heat input relative to the fusion welding processes, still heat input plays a significant role in controlling the joints properties and quality [33]. In this work, the control system in the FSW machine used allows the recording of the spindle torque $T$ (N·m) that can be used with the other FSW parameters such as rotational speed $\omega$ (rpm) and the welding speed $v$ (mm/min) to calculate the heat input. Heat input is defined as the heat energy applied to the workpiece per unit length in the unit of (J/mm). The source of heat generated during FSW is mainly from the friction between the tool and the stirred material and the heat input during FSW can be calculated using Equation (1) [33–35]:

$$\text{Heat Input (J/mm)} = \frac{power}{speed} = \eta \left( \frac{\omega T}{v} \right) \tag{1}$$

$$\text{Where } \omega = \left( \frac{2\pi r}{60} \right) \tag{2}$$

where $T$ is the torque (N·m), $\omega$ is the rotational speed (rpm), $v$ is the linear speed (mm/min) and $\eta$ is the efficiency of heat transfer, ($\eta$ = 0.9) [36,37]. The pseudo heat index is represented by the ratio of the square of the rotational speed to travel speed ($\omega^2/v$). As a function of FSW parameters, it can be considered a simple heat input metric and a well-known method to predict the heat generated during FSW. The maximum temperature highly depends on the rotation tool speed while the heating rate depends on the welding speed at a given tool geometry and plunge depth. The rotation tool speed term is squared because of its significant effect on the heat generated during the process [38]. The pseudo-steady-state welding parameters, calculated heat input and heat index are presented in Table 4.

**Table 4.** Key pseudo-steady-state welding parameters.

| Joint AS-RS | $\omega$ (rpm) | $v$ (mm/min) | $\omega/v$ | Torque (N·m) | HI (J/mm) | Heat Index $\omega^2/v$ |
|---|---|---|---|---|---|---|
| AA5083-AA5754 | 400 | 20 | 20.0 | 91 | 171 | 8000 |
| | 400 | 40 | 10.0 | 117 | 110 | 4000 |
| | 400 | 60 | 6.7 | 116 | 73 | 2666 |
| AA5083-AA5754 | 600 | 20 | 30.0 | 73 | 206 | 18,000 |
| | 600 | 40 | 15.0 | 87 | 123 | 9000 |
| | 600 | 60 | 10.0 | 85 | 80 | 6000 |
| AA5083-AA7020 | 500 | 20 | 25.0 | 87 | 205 | 12,500 |
| | 500 | 40 | 12.5 | 91 | 107 | 6250 |
| | 500 | 80 | 6.3 | 65 | 39 | 3125 |
| AA7020-AA5083 | 500 | 20 | 25.0 | 101 | 238 | 12,500 |
| | 500 | 40 | 12.5 | 85 | 100 | 6250 |
| | 500 | 80 | 6.3 | 104 | 62 | 3125 |

For the joints AA5083/AA5754, Figure 4a shows that the relatively high travel speed (60 mm/min) with low rotational speed (400 rpm) resulted in low $\omega/v$ value (6.66) and consequently low heat input value. Decreasing the welding speed to 40 mm/min for the same rotational speed of 400 rpm in Figure 4a increased the $\omega/v$ value to 10, leading to an increase in the heat input level. In Figure 4a, although the value of is the same ($\omega/v$ = 10) for a travel speed of 60 mm/min and rotational speed of 600 rpm as that in Figure 4a, the increased travel speed of 60 mm/min has showed a more dominant effect than the rotational speed (600 rpm) and resulted in decreasing the HI level. For the other system of joint (AA5083/AA7020; Figure 4b), the heat input can be interpreted in the same manner as explained in Figure 4a. The increased level of $\omega/v$ value (25) in Figure 4b (500 rpm and 20 mm/min) has resulted in obvious increase in the heat input level which reach the value of 261 J/mm. Changing the arrangement of the plates from AA5083/AA7020 to AA7020/AA5083 for the same $\omega/v$ value (6.25) has showed no difference in the power and heat input values, as shown in Figure 4b.

### 3.2. Joint Appearance and Internal Quality

To investigate the joint appearance, the top surfaces of all joints have been visually investigated and pictured. Figures 5 and 6 show the top view of the FSWed AA5083/AA5754 and AA5083/AA7020, respectively. It should be mentioned here that the alloys position at the AS and RS has been ignored in case of the alloys of the same series AA5083/AA5754, while this parameter has been taken into consideration in case of the different series alloys AA5083/AA7020. Figure 5 clearly shows top surfaces free of any surface defects almost at all FSW conditions investigated for this group of alloys except the tool pin breakage at the 600-rpm rotation rate and both 40 and 60 mm/min traverse speeds. The position of the tool breakage is indicated in each top surface by a black arrow. This breakage of the tool pin at the high tool rotation rate and the high tool traverse speed can be attributed to the increase in the applied pressure at the high welding speeds to keep the plunge depth constant. In terms of the flash at the top surface, it is almost minimum under all conditions.

Figure 6 shows the top surface of the FSWed AA5083/AA7020 and AA7020/AA5083 at the same welding conditions for each combination. The surfaces are clearly free of any surface defects expect little flash at the AS especially in case of AA5083/AA7020 that have reduced by reversing the alloys position. Moreover, the reversing alloys position has resulted in tool pin breakage at the highest welding speed condition in the combination AA7020/AA5083. This can be attributed to the resistance of the high strength alloy at the AS especially at the high welding speed of 60 mm/min.

Similar surface features can be visualized in the FSWed joints AA5083/AA7020 and AA7020/AA5083 shown in Figure 6. In some samples, the plasticized flash became clear thick as shown in samples welded at low travel speed (20 and 40 mm/min) where the heat input is higher, and the material is more ductile. This thick flash could be also related to the high applied pressure by the shoulder which leads to excessive penetration of the shoulder in the hot stirred material. At higher travel speed (80 mm/min, Figure 6) the formed flash is thin, discontinuous, and easily dethatched from the FSWed samples. One additional defect is the keyhole formed at the exit of the pin from the material at the end of the welding pass, which is a characteristic defect in the FSWed samples. Finally, it can be said that the welded surface showed a relatively minimum amount of flash which consider as materials loss due to either a higher plunging force or a hotter condition, e.g., a higher rotational speed and/or a lower traverse speed as it will be discussed in studying the FSW heat input.

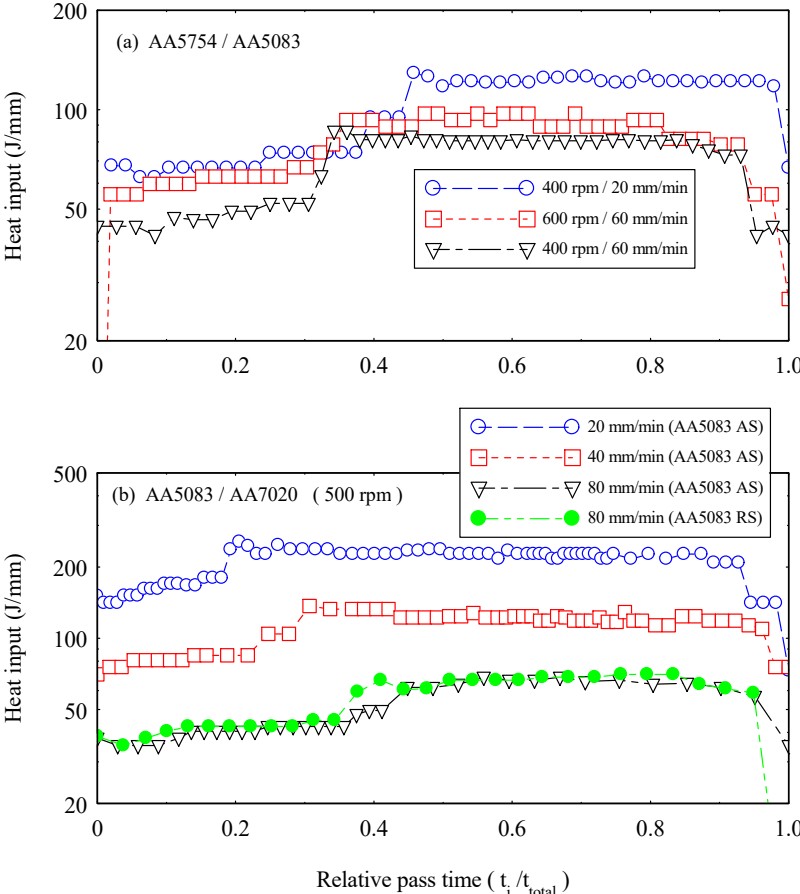

**Figure 4.** Calculated heat input of friction stir welded joints at different rotation and traverse speeds (i.e., at different $\omega/v$ ratios) versus the relative pass time ($t_i/t_{total}$) for FSW for the joints (**a**) AA5083/AA5754 and (**b**) AA5754/AA7020.

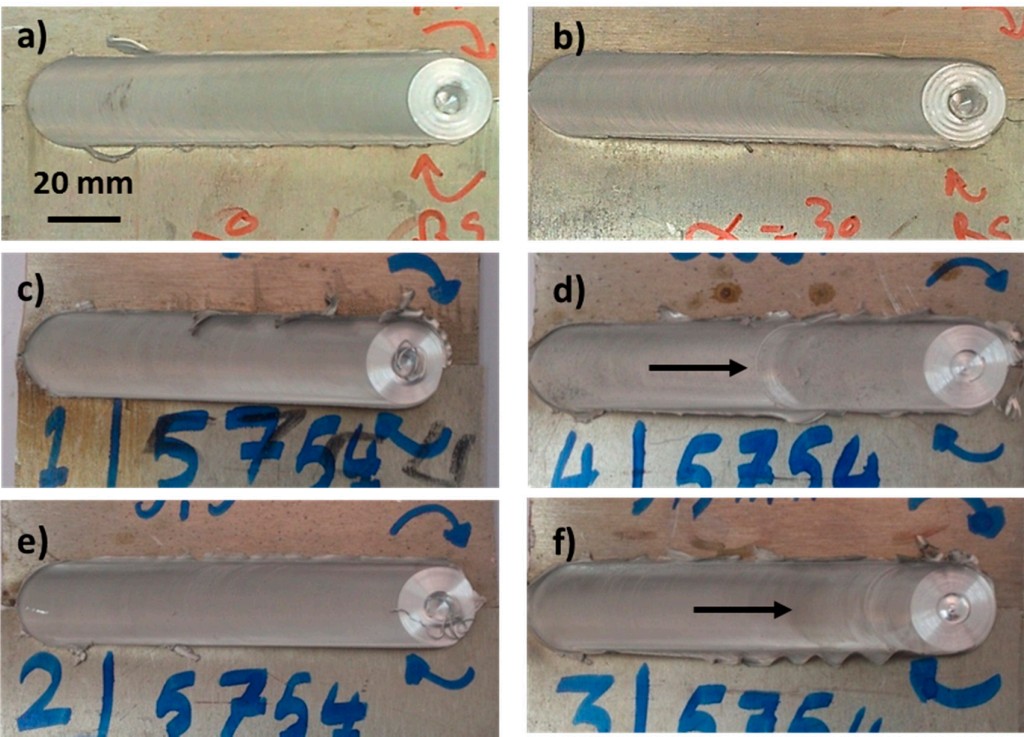

**Figure 5.** Surface appearance of friction stir welded AA5083/AA5754 joints at different rotation and traverse speeds: (**a**) 400 rpm, 20 mm/min, (**b**) 600 rpm, 20 mm/min, (**c**) 400 rpm, 40 mm/min, (**d**) 600 rpm, 40 mm/min, (**e**) 400 rpm, 60 mm/min and (**f**) 600 rpm, 60 mm/min.

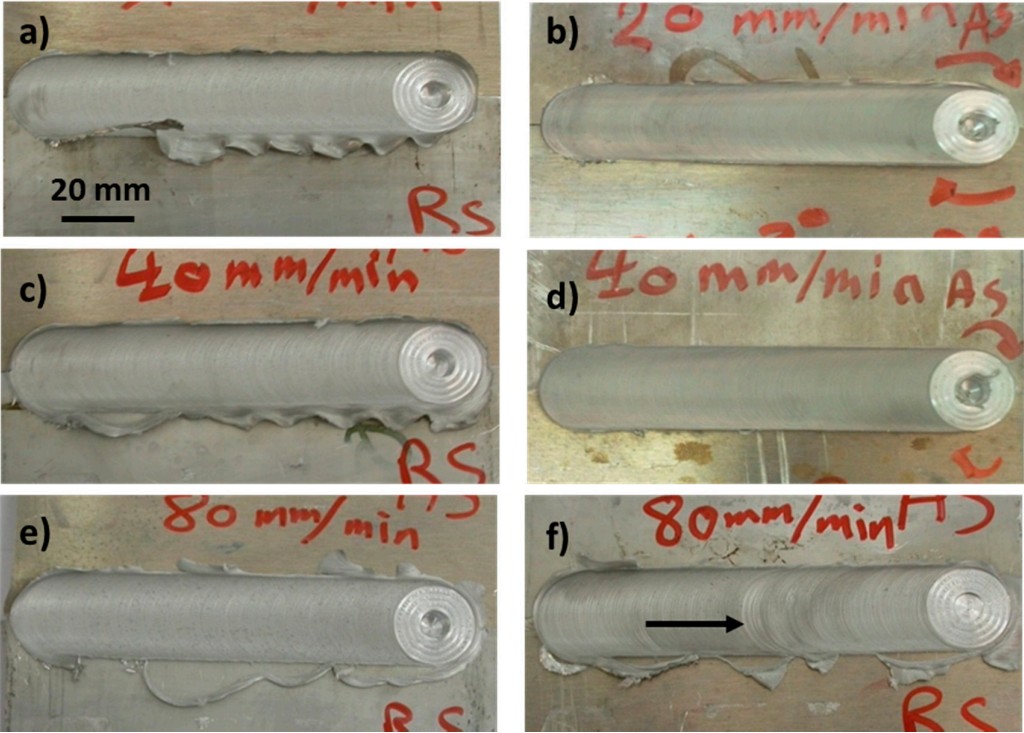

**Figure 6.** Surface appearance of friction stir welded joints AA5083/AA7020 at different rotation and traverse speeds: (**a**) AA5083/AA7020, 500 rpm, 20 mm/min, (**b**) AA7020/AA5083, 500 rpm, 20 mm/min, (**c**) AA5083/AA7020, 500 rpm, 40 mm/min, (**d**) AA7020/AA5083, 500 rpm, 40 mm/min, (**e**) AA5083/AA7020, 500 rpm, 80 mm/min and (**f**) AA7020/AA5083, 500 rpm, 80 mm/min.

Figure 7 shows the transverse cross section macrographs of the polished and etched FSW joints AA5083/AA5754 at rotational speeds of 400 and 600 rpm using traverse speeds of 20, 40 and 60 mm/min. Although several Keller's' reagents have been used at different concentrations to etch the polished sections; the boundaries separating the stirred zone (SZ) and base material are difficult to identify due to the difficulties of etching the AA5XXX Al-alloy group. However, the presented transverse cross section macrographs show defect free joints in two joints out of six made for this combination. The two of the joints welded at 400 rpm and traverse speeds of 40 mm/min and 60 mm/min are completely sound and defect-free, while the joint made at welding speed of 20 mm/min contains a tiny tunnel defect indicated by arrow on the macrograph. This implies that at 400 rpm increasing the welding speed from 20 mm/min to 40 mm/min and 60 mm/min eliminates the tiny tunnel defect. The calculated heat input data above indicates that increasing the welding speed at constant rotation rate results in a decrease of the heat input. The three joints welded at 600 rpm for the same base materials with same arrangement (AS & RS) contain different sizes of tunnel defect from tiny or small to medium size. This indicates that the high heat input will result in tunnel defect, and this can be attributed to the change in the friction condition during the FSW process. There are two friction conditions reported to occur during FSW based on the FSW conditions or based on the heat input namely sticking friction and sliding friction [38]. This would result in some frictional slippage at the shoulder. There could also be instances where the FSW process may alternate between plastic flow and frictional slippage or a stick-slip mode operating at the shoulder. Alternating boundary conditions at the interface may act to destabilize the temperature, which may cause stick-slip oscillations [38]. The AA5083 is reported display poor weldability during FSW due to the strong influence of the plastic properties at high temperatures, on material flow during welding, as well as on contact conditions at the tool workpiece interface [39].

Figure 8 shows the optical macrographs of the transverse cross sections of the FSW joints AA5083/AA7020 and AA7020/AA5083 produced at rotational speed of 500 rpm and traverse speeds of 20, 40 and 80 mm/min. Etching shows the deformation lines of the alloy AA7020 (as it is well known in the AA7XXX Al-alloy series) and the welding zone can be distinguished from the two base plates. The optical macrographs in Figure 8 clearly show that the boundaries between the nugget zone (NG) and the base materials are well defined through the whole thickness. The shape of the NG is wide conical near the top surface due to the large shoulder diameter dominating the stirring and deformation at the top surface. While it is narrow cylindrical near the lower surface due to the small pin diameter dominating the stirring and deformation at the lower surface. A transition can be noted with the conical shape narrowing towards the base. In this transition zone both the shoulder and the pin are contributing to the stirring and deformation. It can be observed that the interface near to the AA7020 is clearer and more distinguished regardless of the AS or RS. This can be due to the effective etching in revealing the micro and macro-features for this alloy in contrast to AA5083. In addition, it can be observed that the interface near to the AA7020 is always free of any defects regardless of the position of the alloy in the AS or RS. Having the AA5083 at the AS has resulted in defect free joint at the welding speed of 20 mm/min, while by increasing the welding speed a very small tunnel defect has occurred at the AS at the welding speed of 40 mm/min and increased in size at 80 mm/min. These tunnel defects can be formed due to the insufficient down force applied during the FSW. This can be attributed to the high resistance of the AA7020 that does not allow the required pressure for the complete consolidation at the applied constant plunge depth. The level of the NG region at the top surface is slightly higher than the level of the base material which supports the scenario of the lack in the applied pressure that causes the tunnel defects. On the other hand, having the AA5083 at the RS has resulted in two defect free joints at the welding speeds of 20 and 80 mm/min. At the welding speed of 40 mm/min, a tunnel defect occurred at the center of the NG near the lower base. This can be attributed to the low applied pressure during the FSW process.

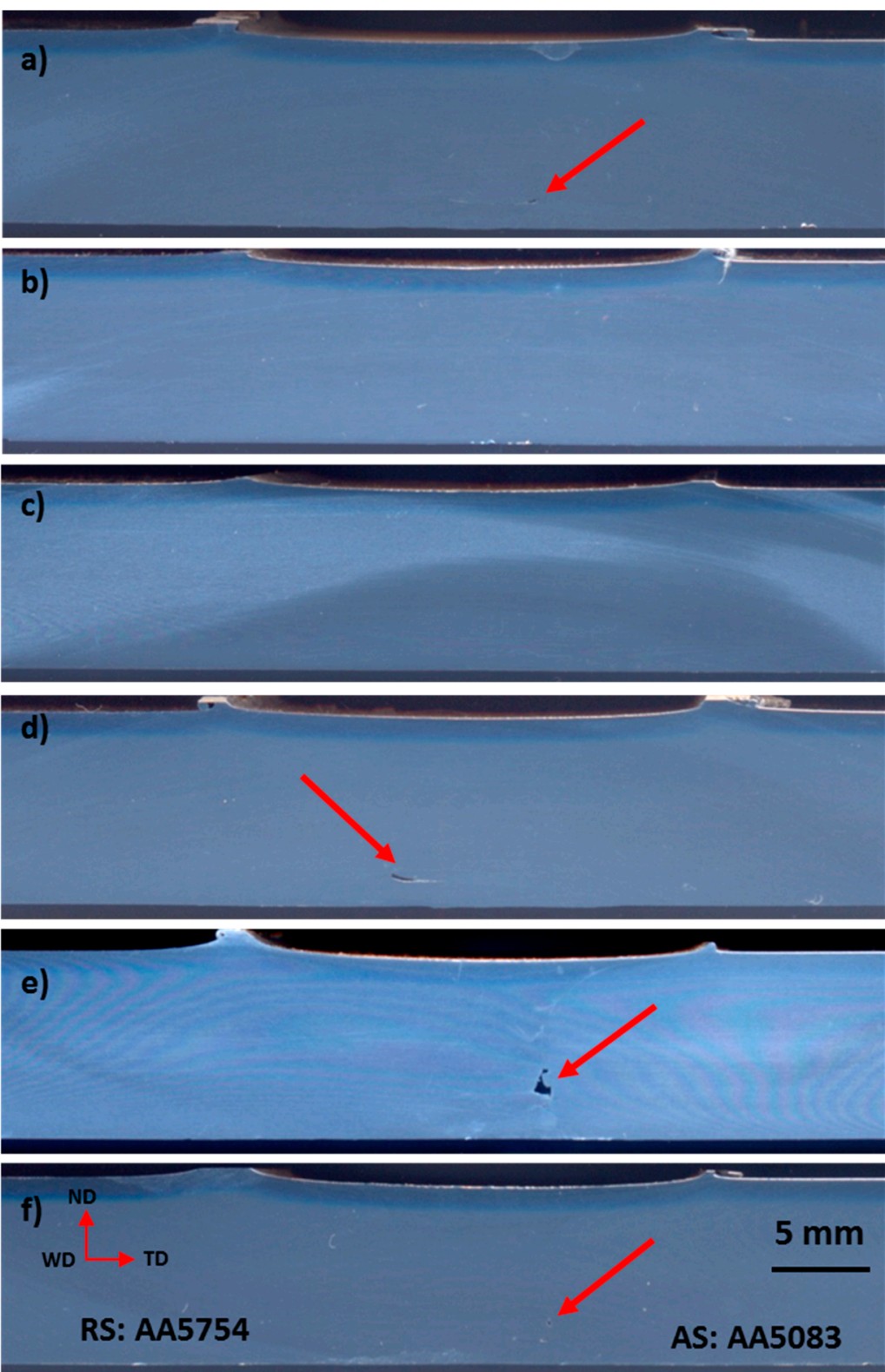

**Figure 7.** Polishing cross-sections of friction stir welded joints AA5083/AA5754 at different rotational and traverse speeds. Arrows refer to the tunnel defects: (**a**) 400 rpm, 20 mm/min, (**b**) 400 rpm, 40 mm/min, (**c**) 400 rpm, 60 mm/min, (**d**) 600 rpm, 20 mm/min, (**e**) 600 rpm, 40 mm/min and (**f**) 600 rpm, 60 mm/min.

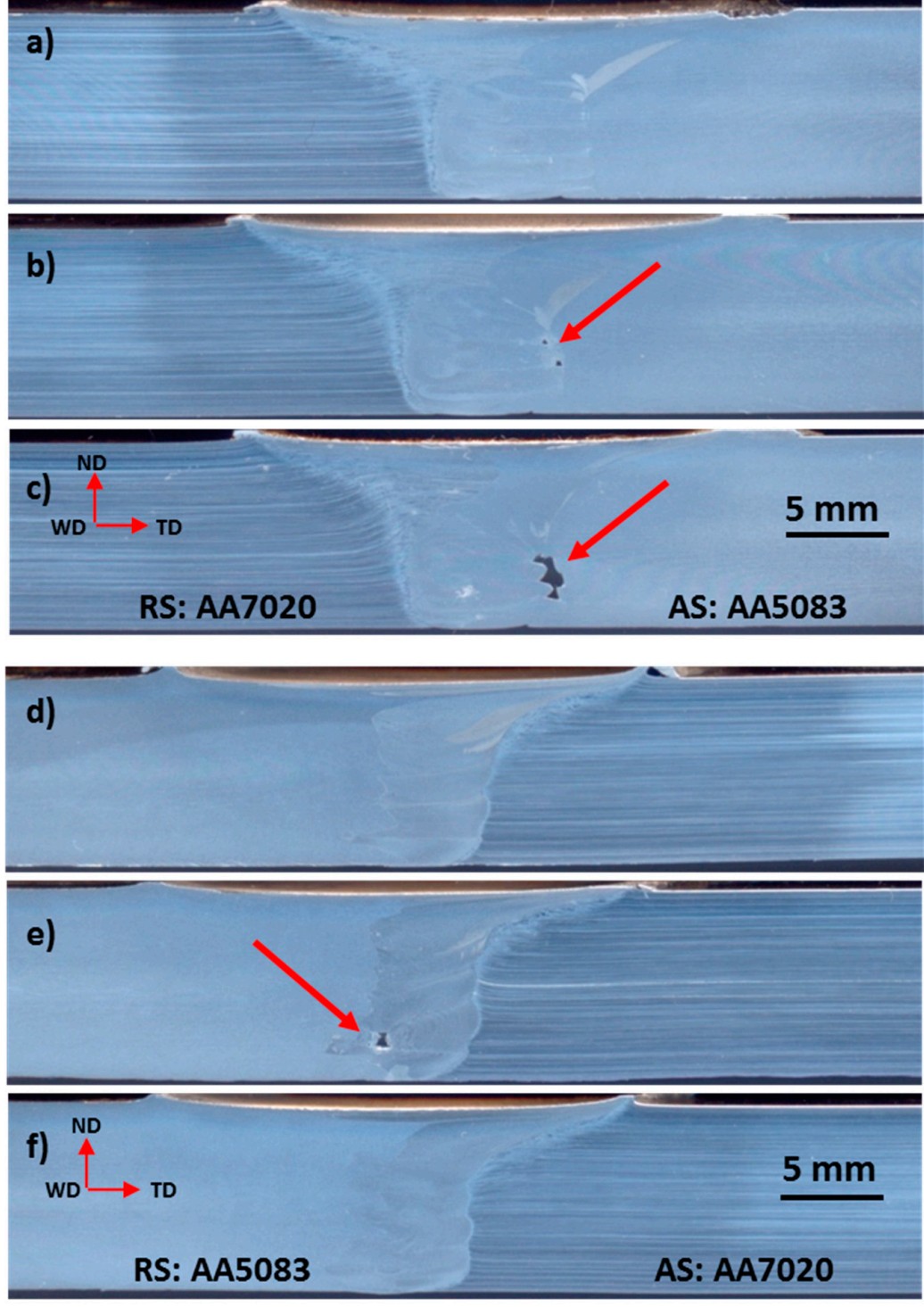

**Figure 8.** Cross sections of friction stir welded joints at different rotational and traverse speeds for AA5083/AA7020 and AA7020/AA5083 after etching: (**a,d**) 500 rpm, 20 mm/min, (**b,e**) 500 rpm, 40 mm/min and (**c,f**) 500 rpm, 80 mm/min.

Table 5 summarizes the defects formed in the produced FSWed joints and the welding conditions. For instance, it is difficult to relate the formed defect to the welding conditions such as the rotational speed ($\omega$), travel speed ($v$) or their combination ($\omega/v$). The general observation is that the internal defects (pin hole or tunnel) are shifted to the side of the softer plate (AA5083) and formed in the lower half of the joint away for the part of the SZ

produced by the rotation of the shoulder. This reflects the positive effect of the pressure excreted by the shoulder on the SZ.

**Table 5.** Visual inspection of the macrostructure of cross section of the FSWed joints and the welding conditions ($\omega$ and $v$).

| Joint | $\omega$ (rpm) | $v$ (mm/min) | $\omega/v$ | Surface Thin Flash and Track Lines | Internal Defect |
|---|---|---|---|---|---|
| AA5083/AA5754 | 400 | 20 | 20 | - | - |
| | 400 | 40 | 10 | thin flash | - |
| | 400 | 60 | 6.66 | - | - |
| | 600 | 20 | 30 | - | pin hole |
| | 600 | 40 | 15 | track lines, thin flash | tunnel |
| | 600 | 60 | 10 | thin flash | - |
| AA5083/AA7020 | 500 | 20 | 25 | thick flash | - |
| | 500 | 40 | 12.5 | thick flash | - |
| | 500 | 80 | 6.25 | thin flash | tunnel |
| AA7020/AA5083 | 500 | 20 | 25 | thin flash | - |
| | 500 | 40 | 12.5 | - | pin hole |
| | 500 | 80 | 6.25 | track lines, thin flash | - |

*3.3. Mechanical Properties*

3.3.1. Macro-Hardness Distribution

Figure 9 shows the hardness profile measured at the midsection of the transverse cross sections of the FSWed joints AA5083/AA5754 at 400 rpm with the different welding speeds in Figure 9a and at 600 rpm with the different welding speeds in Figure 9b. It can be observed that the hardness is reduced in the weld zone with more reduction by increasing the rotation rate from 400 rpm to 600 rpm at each welding speed, and this is mainly due to the increase in the heat input. Moreover, the reduction in the NG hardness is also affected by the increase in the welding speed at the constant rotation rate. It can be noted that the NG hardness is reduced more by decreasing the welding speed from 60 mm/min up to 20 mm/min at the constant rotation rate. In terms of the width of the heat affected zone at each rotation rate, it is reduced by increasing the welding speed. Generally, this hardness reduction in the weld zone is mainly because of the thermal cycle on the strain hardened alloys. The thermal cycle leads to softening of the strain hardened material through the recovery and recrystallization processes that take place during FSW of the aluminum alloys [40].

Figure 10 shows the hardness profile measured at the midsection of the transverse cross sections of the FSWed joints AA7020/AA5083 at 500 rpm with AA5083 at the AS and different welding speeds in Figure 10a and with the AA7020 at the AS and different welding speeds in Figure 10b. It can be observed that both profiles show almost no reduction in the base materials hardness in all the heat affected zone of the two alloys; however, a slight increase in the hardness of the AA7020 alloy can be observed towards the NG center regardless of its position at the AS or RS. This can be attributed to the solid solution strengthening that can occur due to the stirring of the dissimilar alloys [41]. On the other hand, there is a slight decrease in the hardness of the AA5083 towards the NG center regardless of its position at the RS or the AS. This is mainly because of thermal cycle on the softening of the FSWed alloy.

3.3.2. Tensile Properties Analysis

The tensile strength properties of the FSW joints are usually compared with those of the alloy having lower tensile strength [41–43]; here AA5083. For design purposes, the yield stress is used more frequently than the ultimate tensile strength so that the yield stress is determined ($\sigma_{0.2\%}$) for the tested base materials and FSW joints. Relative ultimate tensile strength ($\sigma_{UTS}$ joint/$\sigma_{UTS}$ 5083) and relative yield stress ($\sigma_{0.2\%}$ joint/$\sigma_{0.2\%}$ 5083) of

the produced FSW joints were determined from the tensile stress–strain curves. These relative values (ultimate tensile strength and yield stress) were presented as a function of the welding speeds as shown in Figure 11. Figure 11a summarizes the tensile properties of FSWed joints AA5083/AA5754 at different welding speeds of 20, 40 and 60 mm/min and at rotational speeds of 400 and 600 rpm obtained from the tensile stress strain curves of the FSW samples and related to the tensile properties of the base alloy.

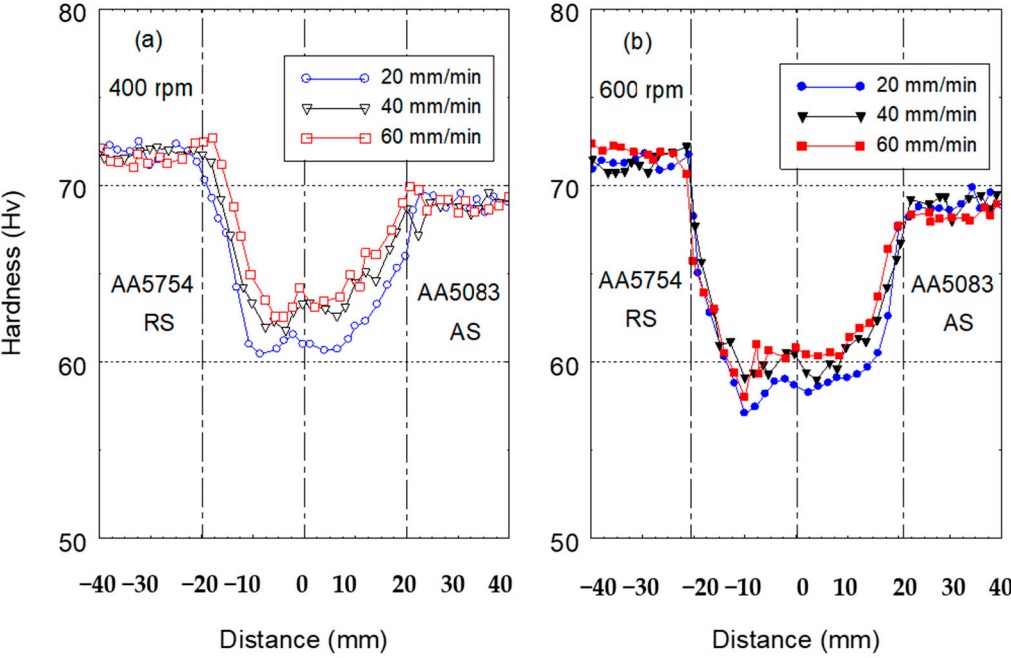

**Figure 9.** Hardness values along FSWed joints AA5083/AA5754 at travel speeds of 20, 40 and 60 mm/min and rotation speeds of (**a**) 400 rpm and (**b**) 600 rpm.

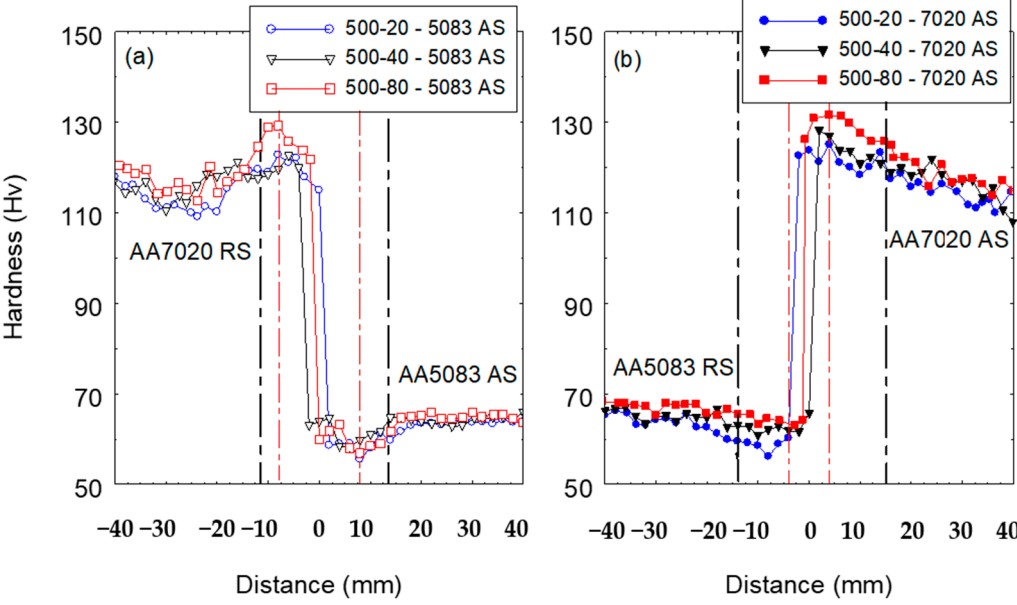

**Figure 10.** Hardness distribution across the joints (**a**) AA5083/AA7020 and (**b**) AA7020/AA5083 welded at travel speeds of 20, 40 and 80 mm/min showing the effect of the joint arrangement.

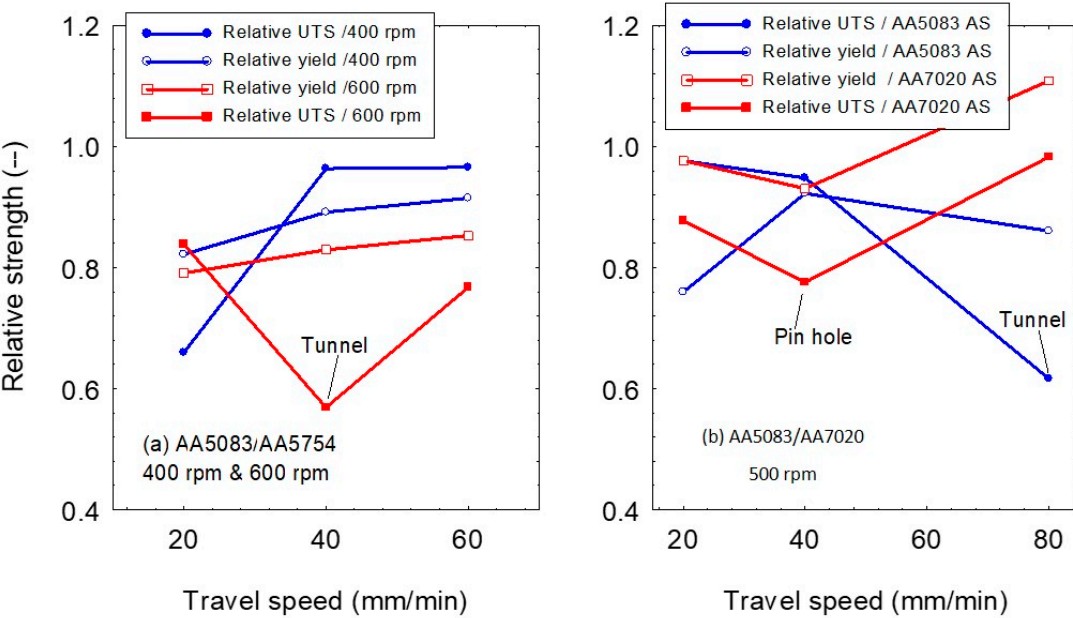

**Figure 11.** Relative tensile strength (σUTS joint/σUTS 5083) and relative yield stress (σ0.2% joint/σ0.2% 5083) of the produced FSW joints against the welding speeds.

As a usual trend, the tensile strength increases with increasing the travel speed and with decreasing the rotational speed, because of the lower generated heat input which permits materials softening. This statement can be supported by the relative yield stress presented in Figure 11a. The relative ultimate tensile strength of FSWed joints is greatly affected by the internal defects in some samples causing early fracture. FSW of the joints AA7020/AA5083 at rotational speed of 500 rpm and travel speed of 80 mm/min has produced joint with ultimate tensile strength comparable with the base alloy AA5083, while the yield stress is even higher than that of the base alloy Figure 11b. Regardless the defect happened at a travel speed of 40 mm/min, clamping the higher strength plate as an AS in friction stir welding of materials with high differences in strength increases the joint strength.

Figure 12 shows macrographs for the fracture positions of the tensile samples of AA5083/AA5754 joints at different welding parameters: (a) 400 rpm–20 mm/min, (b) 400 rpm–40 mm/min, (c) 400 rpm–60 mm/min. and fracture locations of AA5083/AA7020 joints at 500 rpm but different travel speeds and positions; (d) AA7020 AS-20 mm/min, (e) AA7020 AS-40 mm/min and (f) AA7020 AS-80 mm/min. It can be observed that the fracture occurred at the nugget zone in one joint of AA5083/AA5754 (Figure 12a) mainly due to the defect while the fracture occurred away from the NG zone in two joints (Figure 12b,c). In terms of the AA7020/AA5083 joints, it can be observed that the low-speed joint fracture occurred away from the NG (Figure 12d) and the high-speed joints the fracture occurred inside the NG (Figure 12e,f) mainly due to the defects noted. The fracture surface of two samples indicated in Figure 12 are investigated using SEM and EDS analysis. Clearly, it can be observed that from Figure 13 that the fracture mechanism of the dissimilar AA AA5083/AA5754 is ductile mode with very clear dimple features as can be seen in the enlarged micrographs of Figure 13b,c. The inclusions shown in Figure 13d are mostly aluminum and magnesium oxides as detected by the EDX analysis. The lack of adherence of such oxide inclusions with the matrix has accelerated the formation of pores around the inclusions through decohesion between the inclusions and the surrounding material which make the microcracks initiation by the coalescence of the neighboring pores possible. With increasing the applied load, the microcracks grow faster and propagate until the reaching the complete fracture. Figure 14a–d shows the fracture surface SEM micrographs of the dissimilar joint AA5083/AA7020. Clearly the dimple features are dominating, which confirms the ductile fracture mechanism.

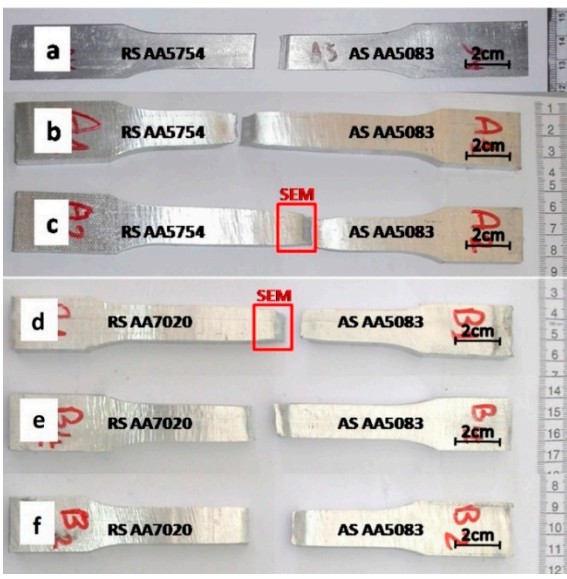

**Figure 12.** Fracture locations of tensile test specimens of AA5083/AA5754 joints at different welding parameters: (**a**) 400 rpm–20 mm/min, (**b**) 400 rpm–40 mm/min and (**c**) 400 rpm–60 mm/min and fracture locations of AA5083/AA7020 joints at 500 rpm but different travel speeds and positions; (**d**) AA7020 AS-20 mm/min, (**e**) AA7020 AS-40 mm/min and (**f**) AA7020 AS-80 mm/min.

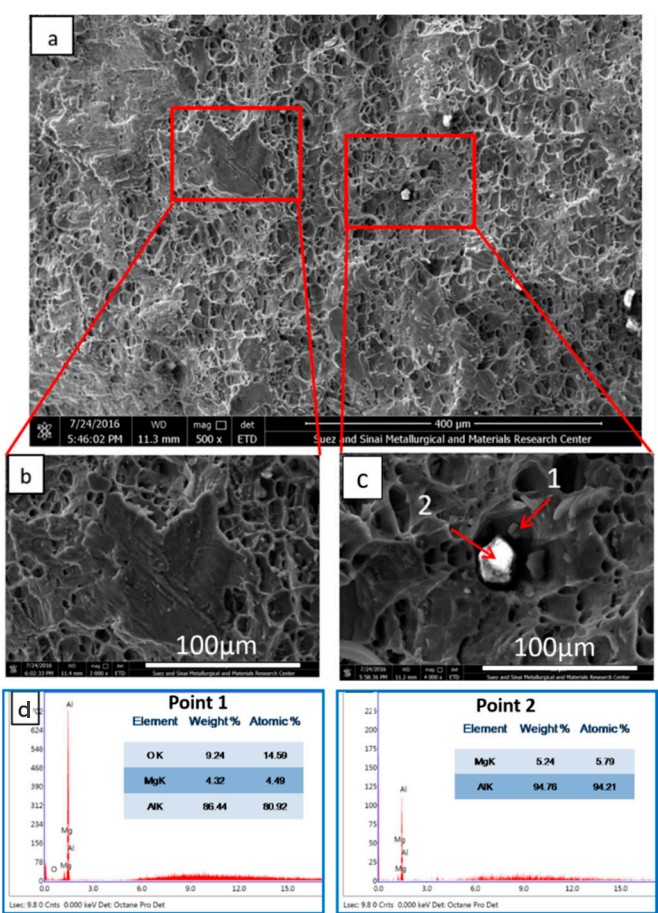

**Figure 13.** SEM micrographs of the fracture surface of tensile test sample of AA5083/AA5754 joint at 400 rpm–60 mm/min, (**a**) mixed fracture mode (**b**) brittle fracture features, (**c**) ductile fracture features, (**d**) EDX spot analysis of the points 1 and 2.

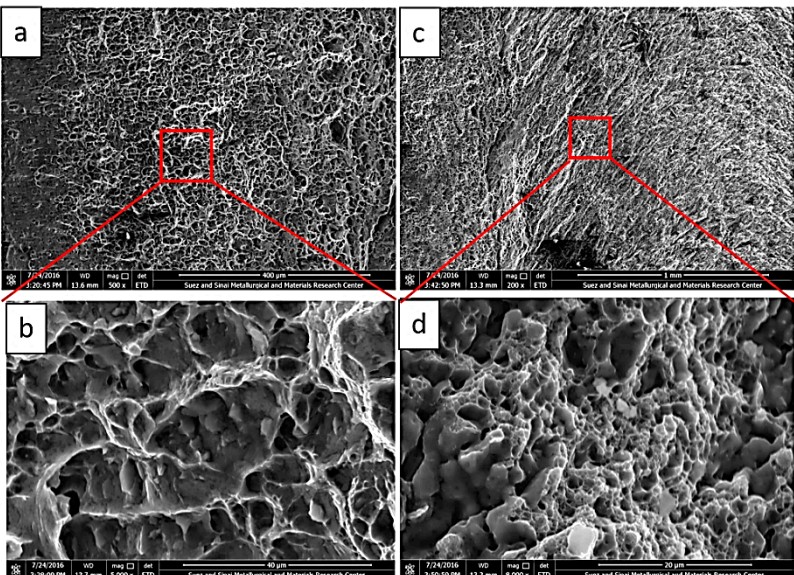

**Figure 14.** (**a**,**c**) SEM images of the fracture surface of tensile test samples of AA7020/AA5083 joint of AA7020 AS, 500 rpm–20 mm/min; (**b**,**d**) show the shape deep dimples.

## 4. Conclusions

In the present study, dissimilar aluminum alloys (AA5083/AA5754 and AA5083/AA7020) were successfully joined by FSW at a wide range tool rotation speed of 300–600 rpm, a traverse welding speed range of 20–80 mm/min and reversing the alloys between the AS and the RS. From the obtained results the following conclusions can be drawn:

— Sound joints are obtained at the low heat input FSW parameters investigated while increasing the heat input results in tunnel defects.
— The hardness profile obtained in the dissimilar AA5083/AA5754 joints is the typical FSW hardness profile of these alloys that reduced in the NG zone due to the loss of the cold deformation strengthening. However, the profile of the dissimilar AA5083/AA7020 showed increase in the hardness in the NG due to the intimate mixing the high strength alloy with the low strength alloy.
— The sound joints in both groups of the dissimilar joints showed very high joint strength with efficiency up to 97 and 98%. Having the high strength alloy at the advancing sides gives high joint strength and efficiency.
— The sound joints showed ductile fracture mechanism with clear dimple features, and significant plastic deformation occurred before fracture. Moreover, the fracture in these joints occurred in the base materials. On the other hand, the joints with tunnel defect showed some features of brittle fracture due the acceleration of the existing crack propagation upon tensile loading.

**Author Contributions:** Conceptualization, S.A., E.A. and M.M.Z.A.; methodology, A.M.A.M., E.A. and M.M.E.-S.S.; validation, S.A., N.A.A. and E.A.; formal analysis, S.A. and M.M.Z.A.; investigation, M.M.E.-S.S., S.A. and E.A.; writing—original draft preparation, S.A. and E.A.; writing—review and editing, M.M.Z.A. and N.A.A.; project administration, M.M.Z.A. and M.M.E.-S.S. All authors have read and agreed to the published version of the manuscript.

**Funding:** This research received no external funding.

**Institutional Review Board Statement:** Not applicable.

**Informed Consent Statement:** Not applicable.

**Data Availability Statement:** The data presented in this study are available on request from the corresponding author. The data are not publicly available due to the extremely large size.

**Acknowledgments:** The authors acknowledge the financial support rendered by the Science and Technology Development Fund (STDF), Ministry of Higher Education and Scientific Research, Egypt.

**Conflicts of Interest:** The authors declare no conflict of interest.

**Abbreviations**

| | |
|---|---|
| $\omega$: | rotational speed, rpm |
| $\eta$: | efficiency of heat transfer, % |
| $\sigma_{0.2\%}$: | 0.2 offset yield stress, MPa |
| $\sigma_{UTS}$: | Ultimate tensile strength, MPa |
| AA: | Aluminum alloy |
| AS: | advancing side |
| BM: | Base Material |
| EDX: | Energy Dispersive X-Ray |
| FSW: | Friction stir welding |
| FSWed: | Friction Stir Welded |
| HAZ: | Heat affected zone |
| HI: | Heat Input, J/mm |
| HP: | Horsepower |
| HRC: | Hardness Rockwell C |
| HV: | Hardness Vickers |
| NG: | Nugget zone |
| Rpm: | Revolution per minute |
| RS: | retreating side |
| SEM: | Scanning electron microscope |
| SZ: | stirred zone |
| $T$: | Torque, N·m |
| TMAZ: | Thermomechanical affected zone |
| $v$: | welding speed, mm/min |
| WN: | Welding nugget |

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
