# Peer review of "Heat Input and Mechanical Properties Investigation of Friction Stir Welded AA5083/AA5754 and AA5083/AA7020"

_metals, doi:10.3390/met11010068_

Round 1

Reviewer 1 Report

The authors present results of heat input and mechanical properties investigation of friction stir welded AA5083/AA5754 and AA5083/AA7020. Considering materials welded, the idea of such a study is new and the manuscript does contain novel results. Based on the obtained results, a detailed description of the influence of the welding parameters on the mechanical properties, heat input and microstructure is presented. The paper is interesting and useful. The presented analysis is logical, and the number of references is sufficient. In general, the presented results are of interest to scientists and engineers. Results of the research are relatively clear, but the manuscript needs a minor revision.  The minor comments are given as follows:

“Materials” section. Please add the table with chemical compositions of the materials welded.

Table 1. Are the welding parameters based on the own research results or based on parameters described in the literature? If parameters were used according to existed literature database, please add a proper reference.

Figure 12. Please add more visible scale bars in Fig. 12 b,c. From which part of the fracture Fig. 12a was taken? Reviewer recommends adding a macro-fractography image of the fracture surface to better overview. The main crack penetrated through SZ, HAZ, TMAZ or base materials zone? Figure 12c and EDX analysis were not discussed in the text. Please add information about the influence of the inclusions on the fracture.

Author Response

Ana Stojkanovic
Assistant Editor, MDPI

Manuscript ID: metals-1054899

Title: “Heat Input and mechanical properties investigation of friction stir welded AA5083/AA5754 and AA5083/AA7020”

Dear Ms. Ana Stojkanovic

On behalf of all the contributing authors, I would like to express our sincere appreciations of your letter and reviewers’ constructive comments concerning our article entitled “Heat Input and mechanical properties investigation of friction stir welded AA5083/AA5754 and AA5083/AA7020” (Manuscript ID: metals-1054899). These comments are all valuable and helpful for improving our article. According to the reviewers’ comments, we have made extensive modifications to our manuscript and supplemented extra data to make our results convincing. In this revised version, changes to our manuscript were all highlighted within the document by using blue colored text. Point-by-point responses to the reviewers are listed below this letter.

Referee: 1

Comment

“Materials” section. Please add the table with chemical compositions of the materials welded.

Response

As pointed out by the learned referee, the nominal chemical compositions of the parent materials are listed in Error! Reference source not found..

Table 1 Nominal chemical composition of aluminum alloys AA5083, AA5754, and AA7020

Alloy

Elements in wt. %

Si

Fe

Cu

Mn

Mg

Zn

Cr

Ti

Al

AA5083

0.40

0.40

0.10

0.4-1.0

4.0-4.9

0.25

0.05-0.25

0.15

Bal.

AA5754

0.40

0.40

0.10

0.50

2.6-3.6

0.20

0.30

<0.15

Bal.

AA7020

0.35

0.40

0.20

0.05-0.50

1.0-1.4

4.50

0.1-0.35

<0.35

Bal.

Comment

Table 1. Are the welding parameters based on the own research results or based on parameters described in the literature? If parameters were used according to existed literature database, please add a proper reference.

Response

The authors would like to thank the referee for the well comment. A described sentence is added to mention how the friction stir welding parameters are chosen, as follows: “The desired welding parameters are based on the ongoing research at the authors laboratory in FSW of the different aluminum alloys of 10mm thick.”

Comment

Figure 12. Please add more visible scale bars in Fig. 12 b,c. From which part of the fracture Fig. 12a was taken? Reviewer recommends adding a macro-fractography image of the fracture surface to better overview. The main crack penetrated through SZ, HAZ, TMAZ or base materials zone? Figure 12c and

Response

As pointed out by the learned referee, figure 12 is added to show the macro-fractographic images of the fracture surface and to estimate where (and why) the fracture occurred.

The old figure 12 becomes Figure 13 and re-sketched to have visible scale bars. Th following paragraph is added to describe the figure 12:

Figure 12 shows macrographs for the fracture positions of the tensile samples of AA5083/AA5754 joints at different welding parameters; a): 400 rpm–20 mm/min, b): 400 rpm–40 mm/min, c): 400 rpm–60 mm/min. and fracture locations of AA5083/AA7020 joints at 500 rpm but different travel speeds and positions; d): AA7020 AS-20 mm/min, e): AA7020 AS-40 mm/min and f): AA7020 AS-80 mm/min. It can be observed that the fracture occurred at the nugget zone in one joint of AA5083/AA5754 (Figure 12 a) mainly due to the defect while the fracture occurred away from the NG zone in two joints (Figure 12 b and c). In terms of the AA7020/AA5083 joints it can be observed that the low-speed joint fracture occurred away from the NG (Figure 12 d) and the high-speed joints the fracture occurred inside the NG (Figure 12 e and f) mainly due to the defects noted.”

Figure 1 Fracture locations of tensile test specimens of AA5083/AA5754 joints at different welding parameters; a): 400 rpm–20 mm/min, b): 400 rpm–40 mm/min, c): 400 rpm–60 mm/min. and fracture locations of AA5083/AA7020 joints at 500 rpm but different travel speeds and positions; d): AA7020 AS-20 mm/min, e): AA7020 AS-40 mm/min and f): AA7020 AS-80 mm/min.

Figure 2 SEM images of the fracture surface of tensile test samples of AA5083/AA5754 joint at 400 rpm - 60 mm/min; mixed fracture mode (b) brittle fracture features, (c) ductile fracture features, d) EDX spot analysis of the points 1 and 2 of AA5083/AA5754 joint welded at 400 rpm - 60 mm/min

The role of the inclusions (in Figure 13) in the fracture has been added in the manuscript:

“The inclusions shown in Figure 13 are mostly aluminum and magnesium oxides as detected by the EDX analysis. The lack of adherence of such oxide inclusions with the matrix has accelerated the formation of pores around the inclusions through decohesion between the inclusions and the surrounding material which make the microcracks initiation by the coalescence of the neighboring pores possible. With increasing the applied load the microcracks grow faster and propagate until the reaching the complete fracture.”

We really appreciate the referee’s valuable comments to improve our manuscript.

Yours sincerely,

Essam Ahmed, on behalf of all co-authors.

Reviewer 2 Report

The reviewer comments of the paper «Heat Input and mechanical properties investigation of friction stir welded AA5083/AA5754 and AA5083/AA7020»

- Reviewer

The authors presented an article «Heat Input and mechanical properties investigation of friction stir welded AA5083/AA5754 and AA5083/AA7020». However, there are several points in the article that require further explanation.

Comment 1:

Overall, the introduction is well written.

At the beginning of the introduction, it is necessary to complete the paragraph. Overall, the introduction is well written. However, explain why the material chosen for research is so important for the study. Provide a paragraph with relevant references for this material. Where is it applied? What are the benefits? What difficulties are observed in friction welding? Clearly identify "white spots". That is, what will be done in this work. Be clear about the novelty of the research.

It is useful to add an article: doi: 10.1155/2019/4156176

At the end of the introduction, summarize what has been done in each section of the article. Provide a clear and understandable purpose of the research you are doing.

Comment 2:

Add a table with the chemical composition of the aluminum alloys AA5083, AA5754, and AA7020. What is the hardness of the workpiece? What is the condition of the workpiece material?

Comment 3:

It will be useful to add a section of Nomenclature in which to sign all the physical quantities and abbreviations encountered in the article. There are many physical quantities in the text and such a section will help to find the description of the necessary element.

Comment 4:

Conclusions.

In addition, it is necessary to more clearly show the novelty of the article and the advantages of the proposed method. What is the difference from previous work in this area? Show practical relevance. Conclusions should reflect the purpose of the article.

Comment 5:

Draw up a list of references according to the requirements of the MDPI publishing house.

The topic of the article is interesting, and relevant. After major changes can an article be considered for publication in the "Metals".

Author Response

Ana Stojkanovic
Assistant Editor, MDPI

Manuscript ID: metals-1054899

Title: “Heat Input and mechanical properties investigation of friction stir welded AA5083/AA5754 and AA5083/AA7020”

Dear Ms. Ana Stojkanovic

On behalf of all the contributing authors, I would like to express our sincere appreciations of your letter and reviewers’ constructive comments concerning our article entitled “Heat Input and mechanical properties investigation of friction stir welded AA5083/AA5754 and AA5083/AA7020” (Manuscript ID: metals-1054899). These comments are all valuable and helpful for improving our article. According to the reviewers’ comments, we have made extensive modifications to our manuscript and supplemented extra data to make our results convincing. In this revised version, changes to our manuscript were all highlighted within the document by using blue colored text. Point-by-point responses to the reviewers are listed below this letter.

Referee: 2

Comment 1:

Overall, the introduction is well written.

-However, explain why the material chosen for research is so important for the study. Provide a paragraph with relevant references for this material. Where is it applied? What are the benefits?

-What difficulties are observed in friction welding?

-Clearly identify "white spots". That is, what will be done in this work. Be clear about the novelty of the research.

-It is useful to add an article: doi: 10.1155/2019/4156176

Response

The authors would like to thank the referee for the valuable comments. The following paragraphs are added:

“AA5754 and AA5083 are aluminum magnesium alloys, and the most prominent features are the high corrosion resistance, and good formability. Thus, they have been extensively used in pressure vessels, tanks, trucks, and shipbuilding [1,2]. AA7020 is a precipitation-hardened aluminum alloy, demonstrating high strength per weight ratio [3].”

“Recently, the friction stir welding (FSW) of dissimilar aluminum alloys combinations has been studied extensively, which proved the potential of the process to join these alloy combinations [4][5][6]. However, improper FSW parameters give rise to the formation of intermetallic compounds, and internal and external defects (e.g., tunnel formation, voids, surface grooves, and flash) [7]. So, the investigation of FSW is very important for obtaining defect-free joints with good mechanical properties.”

-The article: doi: 10.1155/2019/4156176 is added.

Comment 2:

Add a table with the chemical composition of the aluminum alloys AA5083, AA5754, and AA7020. What is the hardness of the workpiece? What is the condition of the workpiece material?

Response

As pointed out by the learned referee, Table 1 is added to present the chemical composition of the aluminum alloys AA5083, AA5754, and AA7020. And the alloys conditions are added into table 2, as follows:

Table 1 Nominal chemical composition of aluminum alloys AA5083, AA5754, and AA7020

Alloy

Elements in wt. %

Si

Fe

Cu

Mn

Mg

Zn

Cr

Ti

Al

AA5083

0.40

0.40

0.10

0.4-1.0

4.0-4.9

0.25

0.05-0.25

0.15

Bal.

AA5754

0.40

0.40

0.10

0.50

2.6-3.6

0.20

0.30

<0.15

Bal.

AA7020

0.35

0.40

0.20

0.05-0.50

1.0-1.4

4.50

0.1-0.35

<0.35

Bal.

Table 2: Mechanical properties of the aluminum alloys AA5083, AA5754, and AA7020

Alloy

Condition

Tensile strength, MPa

Hardness, HV

AA5083-O

Annealed

233

68

AA5754-H14

Strain hardened-1/2 hard

251

74

AA7020-T6

Solution heat treated and artificially aged

364

117

Comment 3:

It will be useful to add a section of Nomenclature in which to sign all the physical quantities and abbreviations encountered in the article. There are many physical quantities in the text and such a section will help to find the description of the necessary element.

Response

Many thanks for referee nice comment. It is added before the introduction section.

List of Nomenclature and Abbreviation

Abbreviation

:

Definition (Expand), unit

w

:

rotational speed, rpm

h

:

efficiency of heat transfer, %

s0.2%

:

0.2 offset yield stress, MPa

sUTS

:

Ultimate tensile strength, MPa

AA

:

Aluminum alloy

AS

:

advancing side

RS

:

retreating side

BM

:

Base material

EDX

:

Energy Dispersive X-Ray

FSW

:

Friction stir welding

HAZ

:

Heat affected zone

HI

:

Heat Input, J/mm

HP

:

Horsepower

HRC

:

Hardness Rockwell C

HV

:

Hardness Vickers

NG

:

Nugget zone

rpm

:

Revolution per minute

SEM

:

Scanning electron microscope

SZ

:

stirred zone

T

:

Torque, N.m

TMAZ

:

Thermomechanical affected zone

v

:

welding speed, mm/min

WN

:

Welding nugget

FSWed

Friction Stir Welded

BM

Base Material

Comment 4:

Conclusions.

In addition, it is necessary to more clearly show the novelty of the article and the advantages of the proposed method. What is the difference from previous work in this area? Show practical relevance. Conclusions should reflect the purpose of the article.

Response

*The conclusion section is rewritten according to the referee’s constructive comments, as follows:

  1. Conclusions

In the present study, dissimilar aluminum alloys (AA5083/AA5754 and AA5083/AA7020) were successfully joined by FSW at wide range tool rotation speed of 300-600 rpm, a traverse welding speed range of 20-80 mm/min and reversing the alloys between the AS and the RS. From the obtained results the following conclusions can be drawn: -

  • Sound joints are obtained at the low heat input FSW parameters investigated while increasing the heat input results in tunnel defects.
  • The hardness profile obtained in the dissimilar AA5083/AA5754 joints is the typical FSW hardness profile of these alloys that reduced in the NG zone due to the loss of the cold deformation strengthening. However, the profile of the dissimilar AA5083/AA7020 showed increase in the hardness in the NG due to the intimate mixing the high strength alloy with the low strength alloy.
  • The sound joints in both groups of the dissimilar joints showed very high joint strength with efficiency up to 97 and 98%. Having the high strength alloy at the advancing sides gives high joint strength and efficiency.
  • The sound joints showed ductile fracture mechanism with clear dimple features mainly and significant plastic deformation occurred before fracture. Also, the fracture in these joints occurred in the base materials. On the other, the joints with tunnel defect showed some features of brittle fracture due the acceleration of the existing crack propagation upon tensile loading.

*The novelty of the article and the advantages of the proposed method can be declared as:

To the author’s knowledge, the FSW of AA5083/AA5754 and AA5083/AA7020 has not been reported in the open literature. This work aims to obtain defect-free welds with good mechanical properties dissimilar aluminum alloys joints which can open new areas of the industrial applications and to benefit from the sustainable advantages such as overall cost reduction, and hybrid properties availability at the two different alloys. FSW can overcomes the fusion welding defects e.g., melting, different kinds of chemical segregations, internal defects, and undesired intermetallic compounds.

Comment 5:

Draw up a list of references according to the requirements of the MDPI publishing house.

Response

Thank you, they were done according to the requirements of the MDPI publishing house.

Round 2

Reviewer 2 Report

The authors have done a good job of improving the article according to the comments.

However, there are two shortcomings:
1. It is better to use "N∙m" for Torque instead of "N.m". Check lines 38, 113, 182, 189, 200.
2. Figure 11 b. 500 rpm overwritten. b) written twice.

After the elimination of these remarks, the article can be accepted for publication.

Author Response

Ana Stojkanovic
Assistant Editor, MDPI

Manuscript ID: metals-1054899

Title: “Heat Input and mechanical properties investigation of friction stir welded AA5083/AA5754 and AA5083/AA7020”

Minor Revision, Round 2

Dear Ms. Ana Stojkanovic

On behalf of all the contributing authors, I would like to express our sincere appreciations of your letter and reviewers’ constructive comments concerning our article entitled “Heat Input and mechanical properties investigation of friction stir welded AA5083/AA5754 and AA5083/AA7020” (Manuscript ID: metals-1054899). These comments are all valuable and helpful for improving our article. According to the reviewers’ comments, we have made extensive modifications to our manuscript and supplemented extra data to make our results convincing. In this revised version, changes to our manuscript were all highlighted within the document by using Track Changes” function. Point-by-point responses to the reviewers are listed below this letter.

Referee:

Comment

On the importance of FSW and its advantages of similar and dissimilar materials joining see some key papers such as:

10.1016/j.matdes.2016.10.038, and 10.1016/j.jmatprotec.2018.07.034, and revise the introduction accordingly.

Response

As pointed out by the learned referee, the references are added.

Comment

What were the conditions of the BMs? Any previous HTT done prior to welding?

Response

The condition of the BMs as given and indicated in Table 2 and there is no heat treatment conducted before or after FSW.Comment

Fig 2 needs a scale.

Response

Scale has been added to the figure as per request.

Comment
English still needs to be polished: "out of sex made for this combination":
should be six.

Response

The article has been read thoroughly and the English has been polished thoroughly.

Comment
Do the authors have zoomed in OM images of the joints?

Response

The microstructures (OM, and SEM) and textures (EBSD) are under investigation in our current work, and it will be submitted to the same journal.

Comment
Other that the manuscript is solid.

Response

Thanks for the positive comment

The authors have done a good job of improving the article according to the comments.

Comment

However, there are two shortcomings:
1. It is better to use "N∙m" for Torque instead of "N.m". Check lines 38, 113, 182, 189, 200.

Response

"N.m" has been replaced with "N∙m" throughout the manuscript.

Comment
2. Figure 11 b. 500 rpm overwritten. b) written twice.

Response

Overwriting of 500 and b has been removed and now corrected.

Comment
After the elimination of these remarks, the article can be accepted for publication.

Response

Thanks for the positive comment

We really appreciate the referee’s valuable comments to improve our manuscript.

Yours sincerely,

Essam Ahmed, on behalf of all co-authors.
